

# The evolution of logic circuits for the purpose of protein contact map prediction

Samuel D. Chapman[1], Christoph Adami[2], Claus O. Wilke[3] and Dukka B KC[1]

[1] Department of Comptuational Science and Engineering, North Carolina A&T State University, Greensboro, NC, USA

[2] Department of Microbiology and Molecular Genetics and Department of Physics and Astronomy, Michigan State University, East Lansing, MI, USA

[3] Department of Integrative Biology, The University of Texas at Austin, Austin, TX, USA

## ABSTRACT

Predicting protein structure from sequence remains a major open problem in protein biochemistry. One component of predicting complete structures is the prediction of inter-residue contact patterns (contact maps). Here, we discuss protein contact map prediction by machine learning. We describe a novel method for contact map prediction that uses the evolution of logic circuits. These logic circuits operate on feature data and output whether or not two amino acids in a protein are in contact or not. We show that such a method is feasible, and in addition that evolution allows the logic circuits to be trained on the dataset in an unbiased manner so that it can be used in both contact map prediction and the selection of relevant features in a dataset.

# INTRODUCTION

Proteins are important biological molecules that perform many functions in an organism. These molecules are composed of a string of amino acids comprised of a 20-letter "alphabet" of amino acids. The sequence of amino acids is referred to as the primary structure of the protein. Beyond this primary structure, proteins are arranged in a higher-order, three-dimensional secondary structure composed of motifs such as alpha-helices and beta sheets. This secondary structure in turn is arranged into a tertiary structure that forms protein domains, which in turn can form a quaternary structure that is composed of multiple protein domains (*McNaught & Wilkinson, 1997*). To a large extent, the two-dimensional and three-dimensional structure of a protein is determined by its amino acid sequence. For example, two cysteine amino acids can form a disulfide bond even if they are separated by a large distance in terms of the sequence (have a large sequence separation) (*Sevier & Kaiser, 2002*). Protein structure in turn greatly influences the function of a protein.

However, it is still fairly time-consuming and expensive to acquire an accurate structure of a protein. Current experimental methods include crystallizing a protein and performing nuclear magnetic resonance (NMR) imaging (*Wuthrich, 1986*) or X-ray crystallography (*Drenth, 2007*). The number of structures that have been determined in

Corresponding author
Dukka B KC, dbkc@ncat.edu

these manners is still quite small—on the order of 100,000 as shown in the latest release of the Protein Data Bank (*Kouranov et al., 2006*) from 2016. This is in contrast to the number of proteins whose primary amino acid sequences have been determined, which is in the tens of millions according to the latest release of RefSeq (*Pruitt, Tatusova & Maglott, 2007*) from 2016. Thus, it is desirable to find faster and cheaper methods for protein structure determination from sequence. One approach is to use computational and machine learning methods to predict protein structure based on available information. These methods are collectively referred to in this paper as protein structure prediction (PSP).

The benefits of using computational methods in PSP are numerous. In addition to providing a less costly and faster method for structural prediction and determination, such methods can also guide and confirm experimental research. Furthermore, structural prediction can help in understanding evolutionary relationships among organisms (*Corbett & Berger, 2004*; *Yoshikawa & Ogasawara, 1991*); aiding in the development of new drugs (*Gaulton et al., 2012*; *Koch & Waldmann, 2005*); and the production of synthetic proteins (*Ho et al., 2001*).

Computational methods used in protein structure prediction are quite diverse. These include clustering methods (*Bolten et al., 2001*), neural networks (*Rost & Sander, 1994*), support vector machines (*Cheng & Baldi, 2007*), and template methods (*Zhang, 2007*). Other examples include those using deep learning (*Lena, Nagata & Baldi, 2012*), which has recently become popular and used in many applications such as image recognition (*Ciresan, Meier & Schmidhuber, 2012*). Computational methods also predict different aspects of structure, including the exact 3D coordinates of the atoms (as in NMR and X-ray crystallography mentioned above), the secondary structures of the amino acids (*Rost & Sander, 1994*), and protein bond angles (*Laskowski, Moss & Thornton, 1993*). There also exist *de novo* or *ab initio* methods, such as ROSETTA (*Simons et al., 1999*) and QUARK (*Xu & Zhang, 2012*), which use only the amino acid sequence to predict protein structure (*Baker & Sali, 2001*).

Another type of structural determination is the amino acid "contact map," which we discuss in this paper (*Cheng & Baldi, 2007*; *Lena, Nagata & Baldi, 2012*). A contact map is the list of amino acids that are in proximity to each other below a distance threshold. In the standard approach to contact map prediction (CMP) by machine learning, the methods are trained on a set of training examples, and after the training periods are tested on a set of unknown testing examples. The training and testing data can encompass any kind of information and class label. For example, in protein contact map prediction, a typical example would be a particular pair of amino acids in a protein. This pair would have many characteristics associated with it (features such as amino acid identity, protein length, and so on), with an accompanying class label (in contact or not in contact) (*Cheng & Baldi, 2007*). The scoring need not be based on class label prediction; for example, in cases where atomic coordinates are predicted, scoring can be based on the predicted distance from the true coordinates (*Cozzetto et al., 2009*).

Computational methods used to elucidate contact maps have included support vector machines (*Cheng & Baldi, 2007*), deep learning (*Lena, Nagata & Baldi, 2012*), integer programming (*Wang & Xu, 2013*), and cascading neural networks (*Ding et al., 2013*). In

particular, in recent years, the use of correlated mutations in predicting contact maps has by been successful in a number of studies *Miyazawa (2013)*, *Jones et al. (2012)* and *Morcos et al. (2011)*. In essence, correlated mutations refer to the fact that evolutionary changes in amino acids can occur together, i.e., in a correlated fashion. For example, in the work of *Miyazawa (2013)*, phylogenetic trees are created from sets of proteins, and partial correlation coefficients are obtained between amino acid sites based on their substitution probabilities and physico-chemical properties. This method was shown to provide direct correlations between sites and achieved high accuracy. Another correlated-mutation method uses a computationally-efficient and high-accuracy version of direct coupling analysis (DCA) *Morcos et al. (2011)*, which is useful when dealing with the problem of secondary correlations (indirect couplings) between amino acids that do not actually interact. To separate direct correlations from these indirect correlations, the DCA method uses what is termed Direct Information (DI) between sites, which is similar to Mutual Information (MI) except that it deals only with direct correlations. This work also uses a mean-field heuristic to speed up the regular DCA computation.

Further direct coupling analysis methods include plmDCA (*Ekeberg et al., 2013*), which uses a pseudolikelihood maximization method based on a 21-state Potts model of the amino acids in a multiple sequence alignment. The DCA method GREMLIN (*Kamisetty, Ovchinnikov & Baker, 2013*) extends the concept of pseudolikelihood maximization, but also uses prior protein information such as secondary structure and amino acid separations using information from the SVMcon program (*Cheng & Baldi, 2007*). Another method, EVfold (*Marks et al., 2011*), uses DCA calculations to produce a set of top-ranked predicted contact pairs, and then uses filtering rules and physical constraint modeling such as simulated annealing to refine the contact models and make final predictions.

Another type of correlated-mutation method, PSICOV *Jones et al. (2012)*, compares a query protein to a multiple sequence alignment and creates a covariance matrix of the amino acids of the proteins at each MSA position. Then, it constructs a sparse inverse covariance matrix, which can help with computation with large matrices. PSICOV has been shown to work well with medium-sized proteins and can help deal with indirect-coupling effects as well.

Neural networks (*Tegge et al., 2009*) and deep learning (*Di Lena, Nagata & Baldi, 2012*), as mentioned above, have also been used in contact map prediction. For example, the program NNcon *Tegge et al. (2009)* uses a recursive 2-D neural network in which the inputs comprise an $L$-by$L$ grid representing the amino acids of a protein, and the outputs are the probabilities of each pair of amino acids being in contact. CMAPpro (*Di Lena, Nagata & Baldi, 2012*) extends this to the idea of deep neural networks, which have multiple layers of nodes that then produce the final predictive outputs. It first uses results from 2-D bidirectional recursive neural networks to obtain approximate contact probabilties and spatial relationships between secondary structure. It then uses an energy-based method to predict approximate contacts between the amino acids of the secondary structures. Then, features from the set of training proteins (spatial features incorporating residue, coarse, and alignment features, along with temporal features) are passed through several layers of
a deep network capable of back-propagation, with each layer having different weights as the others but the same node architecture.

One popular type of machine learning is the use of evolutionary computation (EC), which has also been applied to PSP (*Pedersen & Moult, 1996*). Broadly speaking, this class of machine learning evolves a population of individuals *in silico* that each represent a candidate solution to the problem at hand. The best-performing individuals (i.e., those with the highest score according to a given *fitness function*) tend to perpetuate in the evolutionary process, improving the performance of the method (*Back, Fogel & Michalewicz, 1997*). This class of methods has been used in many applications and takes many forms, in areas as diverse as medical imaging (*Baluja & Simon, 1998*), data mining (*Alcalá-Fdez et al., 2009*), signal processing (*Fogel, 2000*), and artificial life (*Adami, 1998*; *Ray & Hart, 1999*; *Ofria & Wilke, 2004*).

In an evolutionary computation program, the representation of an individual—that is, the digital encoding of the individual—can be varied. These can include tree structures (*Cramer, 1985*), strings of digits (*Edlund et al., 2011*), and even software code itself (*Ofria & Wilke, 2004*). The encoding of these individuals must then be translated into a solution for the problem at hand; for example, in the work of Ofria et al. the digital organisms composed of computer code are scored based on the complexity of logical operations they perform (*Adami, 1998*; *Ofria & Wilke, 2004*).

A particular representation of candidate solutions in evolutionary computation is known as a Markov network. Markov networks are a way of relating mathematical variables to one another. These relations are probabilistic; when the probabilities are either 0.0 or 1.0, they are said to be deterministic. The variables in Markov networks can be arranged such that they are essentially digital logic circuits with deterministic probabilities. That is, they are composed of logic gates that accept binary inputs (0 or 1) and produce binary outputs. Markov networks have in recent years become a tool in areas such as the navigational control (*Edlund et al., 2011*), active categorical perception (*Marstaller, Hintze & Adami, 2013*) and machine learning in image recognition (*Chapman et al., 2013*).

This paper examines the use of evolving, deterministic Markov networks toward the problem of contact map prediction. In this work, we evolve Markov networks on a training set comprised of pairs of amino acids, and at the end of the evolution test the best networks on a testing set. The class labels of the examples are whether or not the pairs are in contact or not in contact, and the examples have a mixture of 688 binary and decimal features that describe them. The training and testing data are each comprised of several hundred thousand examples containing both negative and positive contacts. The dataset used is taken from SVMcon, a paper that uses support vector machines for protein contact map prediction (*Cheng & Baldi, 2007*). In our work, the features are used as inputs to Markov networks (logic circuits), and the outputs are the class labels. Our results show that evolving logic circuits to predict protein contact maps is a viable alterinative. To our knowledge, this is the first time that Markov networks have been used in machine learning on a bioinformatics problem. Thus, it is a feasible avenue for study and has the possibility of being quite useful for this problem. Of course, as there is continual improvement in the

fields of evolutionary computation and Markov networks, this study should not be seen as representing the "last word" on the capacity of this method to tackle this problem.

Although contact map prediction has improved over the years, there are still a number of challenges remaining. For example, it is unlikely that the entire feature set is useful for classification, a fact mentioned in the SVMcon paper (*Cheng & Baldi, 2007*). Indeed, it is extremely difficult to know beforehand whether a subset of features performs better on the task, and it may be that the only way to determine this for sure is to manually take out features and re-run the method, an approach that is impractical. In addition, if it is true that a subset of features may be better than a full set, then it follows that it is difficult to know if the addition of more features may help to solve the problem.

Evolutionary computation has a number of benefits and addresses some of the concerns outlined above. For example, once the appropriate fitness function has been selected (along with the evolutionary parameters such as mutation rate and number of individuals in the population) the program attempts to evolve a most-fit individual according to the fitness function and fitness landscape. There is no "correct" information that the EC system is told to use, but rather only the information that produces the best outcome through evolution. Thus, when combined with Markov networks, an EC system can *discover relevant features* in an unbiased manner. This is demonstrated by observing which features Markov networks evolved to use in their structures—if a feature was used, this indicates that it helped in increasing fitness and was therefore important.

## MATERIALS AND METHODS

### Description of Markov networks and their evolution

Markov networks are a set of probabilistically interacting state variables (*Koller & Friedman, 2009*). The sets of state variables in a Markov networks are often arranged into input state variables, output state variables, and "hidden state" variables that are "inside" the network and can serve as memory and can be used for processing. This arrangement can be used to represent many processes, such as robotic controllers (*Edlund et al., 2011*) (e.g., with sensory inputs and movement outputs) or in image classification (*Chapman et al., 2013*) (e.g., pixels as inputs and image classes as ouptuts). In this work, the tables that determine the network updates are deterministic rather than probabilisitic, so that Markov networks represent regular logic circuits.

An example of a 2-input, 2-output gate is shown in Table 1. Here, *a* and *b* represent the binary inputs and *c* and *d* represent the binary outputs.

In this study, Markov networks (logic circuits) are created through an evolutionary process. Therefore, the networks themselves must be encoded symbolically into a "genome." This is accomplished by representing Markov networks as a string of integers (bytes), the length of which can be arbitrary but is limited to 40,000 integers in our work. The genome of an individual Markov network is comprised of a set of "genes," which each specify one logic gate. Each gene that represents a logic gate begins with an integer that denotes the start of the gene, and the gene itself specifies where the specific inputs and outputs are located, as well as the logic of the gate. This information is sufficient to describe

**Table 1  Gate logic table.** The characteristic logic table for a deterministic gate with two inputs and two outputs.

| Inputs a b | Outputs c d |
|---|---|
| 1 1 | 0 1 |
| 0 1 | 0 0 |
| 1 0 | 1 1 |
| 0 0 | 0 1 |

the entire Markov network. Any other characteristics of the candidate solutions, such as the maximum genome size, the rates of mutation, or the number of inputs and outputs to a gate, is specified in a configuration file.

The evolution of Markov networks in our study follows simple evolutionary operators that act on the genome of an individual. Evolutionary operators that can act upon the genes include point mutation (which changes a single byte) or gene duplication and deletion, which duplicate or delete an entire gene, respectively. The rates of each of these are specified in the configuration file. In our work, if two or more genes write into the same output, the the final output is OR'd together. More details of the general makeup of Markov networks are provided in the Supplementary Information of *Edlund et al. (2011)*. The evolutionary algorithm code used is based off the work by *Goldsby et al. (2014)*, and is a version of the code available at https://github.com/dknoester/ealib.

## Dataset description

Our work centers on protein structure prediction (PSP); specifically, we investigate how protein feature data can be used to predict the amino acid contact map. In order to do this, a dataset of protein sequences and protein sequence features must be prepared. We choose to use the feature dataset provided in *Cheng & Baldi (2007)*. That paper described the creation of a contact map predictor based on support vector machines, SVMcon. Their dataset included several hundred proteins divided into a training dataset used to train the SVM (485 proteins) and a testing dataset (48 proteins). The proteins used are taken from the work of *Pollastri & Baldi (2002)*, and the sequence identity between any two proteins is no more than 25 percent. In addition, all proteins have uninterrupted backbones and are under 200 amino acids in length.

Each protein was used to provide a large number of amino acid contact examples based on the many possible amino acid pairs. The sequence separation threshold for pairs was six or greater, and the threshold for contact (based on distance between the alpha-carbons of the amino acids) was eight Angstroms or greater.

Each pair of amino acids in the training or testing set represented a possible contact pair, i.e., each pair was either in contact or not in contact. The training dataset contained 267,702 contact pairs, reduced from a set of several million. This reduction was done in order to make the dataset more tractable and also to increase the proportion of positive contacts; the final training dataset had 94,110 positive contacts, a proportion of roughly one-third. The reason the proportion of positive contacts was increased in the training dataset was

to train the SVM to better recognize these examples. The testing dataset, composed of 377,797 examples, used the full proportion of contact examples and contained 10,498 positive contacts.

Each training and testing example had 688 features associated with it. These features included many types of data related to the sequences, such as mutual information, length of sequences, and secondary structure, to name a few. There were 145 binary features (0 or 1) and 543 decimal features (capable of being any number). Many of these features were based on a sequence alignment of the proteins, which allowed all sequences to be compared according to a standardized length. PSI-BLAST *Altschul et al. (1997)* was used to compare each sequence to the NCBI non-redundant database *Pruitt, Tatusova & Maglott (2007)* and to create the multiple sequence alignments with their profiles. The majority of features were based on sliding windows centered around the amino acids in each pair and a window centered halfway between the amino acids. There were nine window positions each for each amino acid in the pair (including the amino acid in question) and five positions for the central segment. For each of these 23 positions, there were 27 features giving the entropy (one feature), secondary structure (three features), solvent accessibility (two features) and the amino acid profiles for those positions (21 features). Thus, the window features comprised $23 * 27 = 621$ out of 688 features. It should be noted that out of the 688 features, only those features associated with secondary structure and solvent accessibility could not be calculated precisely from the sequence alone. Secondary structure and solvent accessibility were predicted using other software (*Cheng et al., 2005*). Table 2 outlines the 688 features used. A more-detailed description of the dataset and features used can be found in the SVMcon paper (*Cheng & Baldi, 2007*).

## Data encoding

Markov networks take binary input. Since many of the features are continuous, it is necessary to develop an input encoding that maps to the features. There are several methods that were tried.

The methods that were tried were based on splitting continuous features into a set number of bins. For each type of encoding, a "split" number was given (in our case, there were three splits used—four, 10, or 16). Each continuous feature was divided into a split number of bins based on the complete range of that particular feature (in the training set). For example, if using a split of 10, and with a feature with a range from 0.0 to 2.0, the first bin would be from [0.0–0.20] the second from [0.2–0.4], and so on. If the testing set had a range that was out of bounds of the training set, the lowest and/or highest bin from the training set was used. In order to make all feature representations equal, each binary feature had the same number of inputs as a continuous feature; for example, if the split was 4, then each binary input used four bits, which were either all 0's or all 1's.

Two additional types of encoding were based on the fact that even-numbered splits can be expressed in terms of base 2. For example, a split of four bins could be compressed into two bits by using the binary values 11, 10, 01, and 00, and a split of 16 bins could be compressed into four bits along the same principle. These two binary splits used were: First, a split of 16 across four bits, and a split of four across two bits.

**Table 2 Dataset features.** A description of the features of the dataset used in this study.

| Feature (s) | Number | Binary | Description |
| --- | --- | --- | --- |
| Cosine similarity | 1 | No | Cosine similarity of amino acid profiles in positions $i$ and $j$. |
| Correlation measure | 1 | No | Correlation measure of amino acid profiles in positions $i$ and $j$. |
| Mutual information | 1 | No | Mutual information of amino acid profiles in positions $i$ and $j$. |
| Amino acid types | 10 | Yes | Gives all types of amino acid in pair among nonpolar, polar, acidic, and basic. |
| Levitt's contact potential | 1 | No | Amino acid pair energy measure. |
| Jernigan's pairwise potential | 1 | No | Amino acid pair energy measure. |
| Braun's pairwise potential | 1 | No | Amino acid pair energy measure. |
| MSA amino acid profiles | 483 | No | Profile of each of the 20 amino acids, plus gap, in the 18 sliding window positions and five central segment positions. |
| MSA entropy | 23 | No | Profile entropy of each of the 18 sliding window positions and five central segment positions. |
| Solvent accessibility | 46 | Yes | Solvent accessibility of the amino acid (buried or exposed) of each of the 18 sliding window positions and five central segment positions. |
| Secondary structure | 69 | Yes | Secondary structure of the amino acid (helix, sheet, or coil) of each of the 18 sliding window positions and five central segment positions. |
| Central segment amino acid compositions | 21 | No | Overall proportions of each of the 20 amino acids, plus gap, across all central segments. |
| Central segment secondary structure compositions | 3 | No | Overall proportion of the three secondary structures across the central segments. |
| Central segment solvent accessibility compositions | 2 | No | Overall proportion of the two solvent accessibilities across the central segments. |
| Amino acid sequence separation | 16 | Yes | Amino acid sequence separation using bins <6, 6, 7, 8, 9, 10, 11, 12, 13, 14, <19, <24, ≤29, ≤39, ≤49, and ≥50. |
| Protein secondary structure composition | 3 | No | Overall secondary structure composition of the protein of the contact pair. |
| Protein length | 4 | Yes | Length of the protein of the contact pair using bins ≤50, ≤100, ≤150, >150. |
| Protein solvent accessibility composition | 2 | No | Overall solvent accessibility composition of the protein of the contact pair. |

We chose these five particular splits for a number of reasons. Based on the general layout of the data, we decided that the lowest split should be four as anything less would be too granular on feature data that tended to have at least four, and often more, distinct values. Second, the highest split of 16 was chosen as a reasonable upper limit, and 10 was chosen as the midpoint between the lowest and highest splits. There is no theory suggesting which split to use, but in practice Markov networks can sometimes have difficulty evolving to find the correct inputs if there are too many of them, which slows down the clock time of the simulation itself (*Chapman et al., 2013*). The number of inputs for the split of four was $4*688 = 2,752$, the number for the split of 10 was $10*688 = 6,880$, and the number for the

split of 16 was 16*688 = 11,008. Due to time limitations, it was impossible to perform an exhaustive parameter sweep of all the splits, even from two to 16.

## Evolution of Markov networks on the dataset

The training (evolution) of Markov networks is achieved by the following steps. We first create an initial population of random networks. We then evolve these networks over a number of updates on a subset of the training dataset. In our case, we ran the evolution over 100,000 updates. For every 25,000 updates, we evolved the networks on a different set of 50,000 training examples. The sets of training examples were separate; thus, we used a total of 200,000 training examples. We used different sets of training examples for two reasons. First, it forced Markov networks to evolve on different examples over time instead of focusing only on one dataset. Second, because of time limitations, it was not possible to evolve the networks on a greater number of examples.

During the runs, the networks "learn" (through evolution) to differentiate between positive and negative contacts based on their features. In each update, the networks are tested on the training set and a fitness is assigned to them according to a fitness function. Individuals with the highest fitness tend to survive in the population and reproduce; these "children" undergo mutation, and in the process, may do better than their parents at recognizing contacts. Mutation can affect almost any part of the networks, including the number of features used as input, the number of gates, and the gate logic. In the end, there are two binary outputs provided: an output for a negative contact decision, and one for a positive contact decision. Note that it is possible for a Markov network to give both answers, signifying both a positive and negative contact decision, but in practice, this does not happen very often.

In our work, we have tried a number of fitness functions, and so far accuracy has proven to be the best in terms of the final scoring method, described below. Accuracy is defined as TP + TN / P + N, where TP is the number of true positive (correctly-predicted) guesses and TN is the number of true negative (correctly-predicted) guesses in the dataset. Here, accuracy is calculated separately for both positive contacts and negative contacts. Also, a reward for output alone is also included for each accuracy to encourage "empty" networks to evolve to produce outputs. The vector magnitude of these results is combined as shown in (Eq. 1), and therefore the maximum fitness is the square root of 8.0. Calculating the square root is done in order to "smooth" the resulting fitness values. At each update, every newly-produced individual Markov network that has not been tested on the training set is tested according to the fitness function.

$$f = \sqrt{(\text{acc}_{\text{pos}} + \text{out}_{\text{pos}})^2 + (\text{acc}_{\text{neg}} + \text{out}_{\text{neg}})^2}, \tag{1}$$

At the end of an evolutionary run, the best Markov network in a population is tested on the testing set, which it has not seen before, and performance is assessed according to specificity, sensitivity, and Fmax. Specificity is the number of correctly-predicted positive contacts divided by the number of total predicted positive contacts (thus, the false positive rate is 1-specificity). Sensitivity, also known as the true positive rate, is the number of correctly-predicted positive contacts over the total number of positive contacts. Because it

**Table 3  Parameters.** Parameters for the evolutionary algorithm.

| Parameter | Value |
| --- | --- |
| Updates | 100,000 |
| Population size | 500 |
| Starting gates | 100 |
| Inputs per gate | 4 |
| Outputs per gate | 4 |
| Gene duplication rate per update | 0.05 |
| Gene deletion rate per update | 0.05 |
| Site mutation rate per update | 0.001 |

is easy to get a specificity of 1.0 by simply giving one very good guess, and a sensitivity of 1.0 by simply guessing all contacts to be positive, Fmax is also used, which combines the two measures and is shown in (Eq. 2). The maximum Fmax possible is 0.5.

$$Fmax = \frac{specificity * sensitivity}{specificity + sensitivity},$$

(2)

Note that negative contacts are not considered in the three performance measures (even though they are used in the fitness function), since positive contacts are of more interest and more difficult to determine in contact map prediction. Finally, even though Fmax is used as the scoring measure, we determined that it was inferior to accuracy as a fitness function.

In order to achieve better performance, committees of Markov networks are assembled from the highest-fitness individuals from each of the 60 evolutionary runs. These 60 Markov networks are tested on the testing set, and for each testing example, the sum of all 60 negative contact answers is compared to the sum of all 60 positive contact answers. The final answer is the answer with the highest number of votes (ties are broken randomly). It should be noted that because of the nature of Markov networks as a classifier tool, it is not possible using our current system to assign a traditional confidence for the guesses of Markov networks. That is, the guess of a particular network does not produce a numerical confidence as to the correctness of the answer (like with a suport vector machine learner such as SVMcon), and is simply a yes or no. However, it is possible to emulate confidence scores by taking the number of committee members for each answer that give a positive guess; for example, if 60 committee members agree that an example is a positive contact, one can treat that as an answer that has higher confidence than if only 45 committee members guessed that it was a positive contact. This technique is used later and described in more detail when comparing our method to others such as SVMcon, where it was necessary to fix the number of positive guesses to achieve more-accurate comparisons.

The list of parameters for the evolutionary runs is given in Table 3.

## RESULTS

### Performance on the dataset

Markov networks were evolved using five different input data encodings (treatments), described in the Methods section. For each encoding, 60 different populations were run

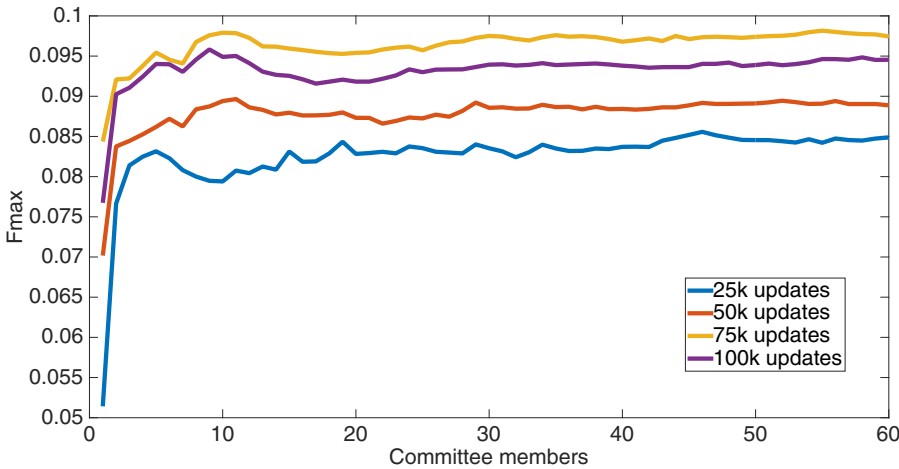

**Figure 1** **Treatment results.** Results for the split of 10 with 10 bits per feature (6,880 total bits). The highest Fmax at 60 committee members is at 75 k, with an Fmax of 0.098.

and the best individuals from each population (according to the fitness function) were tested on each example from the testing set and their answers combined. The evolution continued for 100 k updates, and at intervals of 25 k updates, results on the test set for committee sizes up to 60 were recorded. Figures 1–5 show the results for the treatments that used splits of 10 with 10 bits per feature, splits of four with four bits per feature, splits of 16 with 16 bits per feature, splits of four with two bits per feature, and splits of 16 with four bits per feature, respectively. Even though the treatments ended at 100 k updates, the best results for all treatments were from 75 k updates; this is probably due to overfitting of Markov networks to the training set in their evolution. Also, the improvement in performance of each treatment at the four update intervals tended to level out after around 20 committee members and does not tend to improve with more. This is an indication that increasing the number of committee members would not help the performance for this treatment.

Figure 6 shows the Fmax results at 75 k updates for all treatments. It is interesting to note that the two best treatments were the two that compressed the number of bits per feature according to a base-2 compression: the treatment that used a split of 16 with four bits/feature had a final Fmax of 0.103 and the one with a split of four with two bits/feature had a final Fmax of 0.102. The splits of 10 with 10 bits per feature, 16 with 16 bits per feature, and 4 with four bits per feature had Fmax results of 0.098, 0.097, and 0.097, respectively.

All of the treatments did significantly better than random. Under random guessing that guessed 50 percent of the examples to be positive contacts, the Fmax would be 0.026. Under random guessing that used the proportion of positive contacts from the training set (35.155 percent), the Fmax would also be 0.026.

We now take a detailed look at the treatment that used a split of four with two bits/feature. This treatment was chosen for two reasons. First, its Fmax performance is very close to the best (Fmax of 0.102 vs. Fmax of 0.103 obtained by the encoding using a split of 16 with

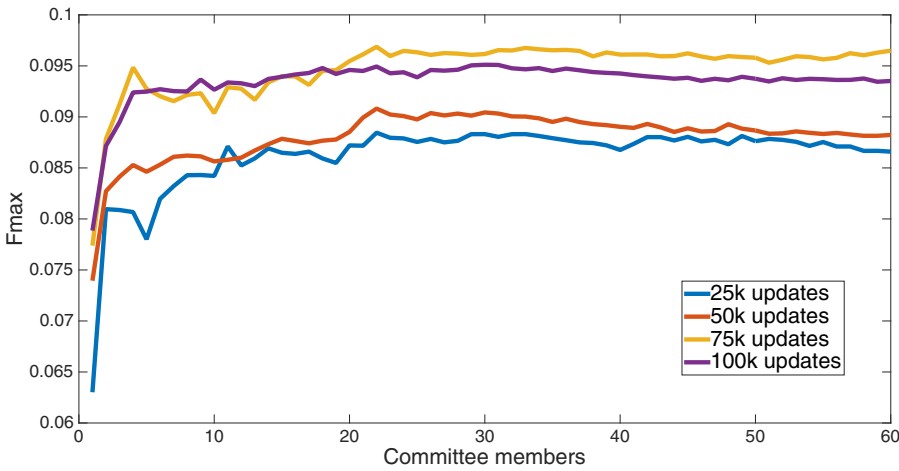

**Figure 2  Treatment results.** Results for the split of 16 with 16 bits per feature (11,008 total bits). The highest Fmax at 60 committee members is at 75 k, with n Fmax of 0.097.

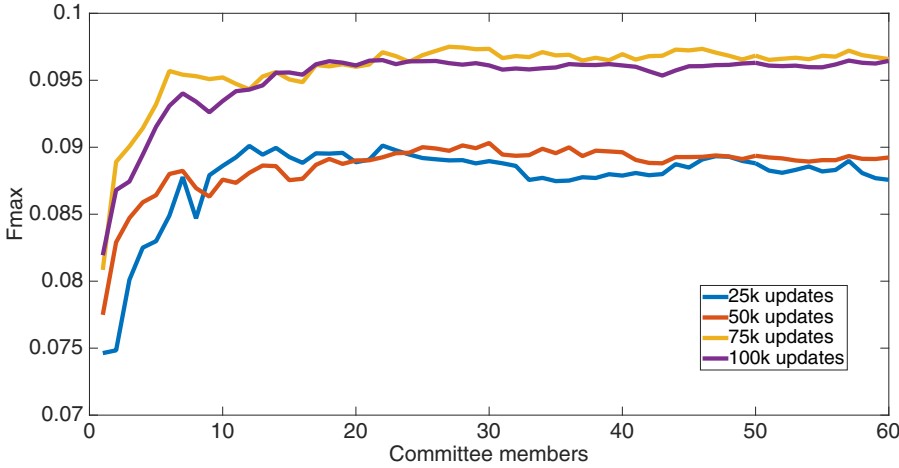

**Figure 3  Treatment results.** Results for the split of four with four bits per feature (2,752 total bits). The highest Fmax at 60 committee members is at 75 k, with an Fmax of 0.097.

four bits/feature); second, it is the simplest type of encoding, using only two bits/feature, making it simpler to analyze and describe. Figure 7 shows the specificity and sensitivity values over the committee sizes at 75 k updates for the split of four with two bits per feature treatment. The specificity at 60 committee members was 0.14, and the sensitivity was 0.35. As one can see, even if sensitivity goes down, specificity can go up, leading to a higher Fmax value.

We also compare our results from the split of four, two bits/feature encoding to the results from the SVMcon paper by *Cheng & Baldi (2007)*. In the SVMcon work, for each protein and sequence separation length, they provided the top-$L$ highest-confidence positive-contact guesses, where $L$ represented the length of the protein. The three different sequence separations they used were $\geq 6$ (i.e., all separations), $\geq 12$, and $\geq 24$. Thus, for

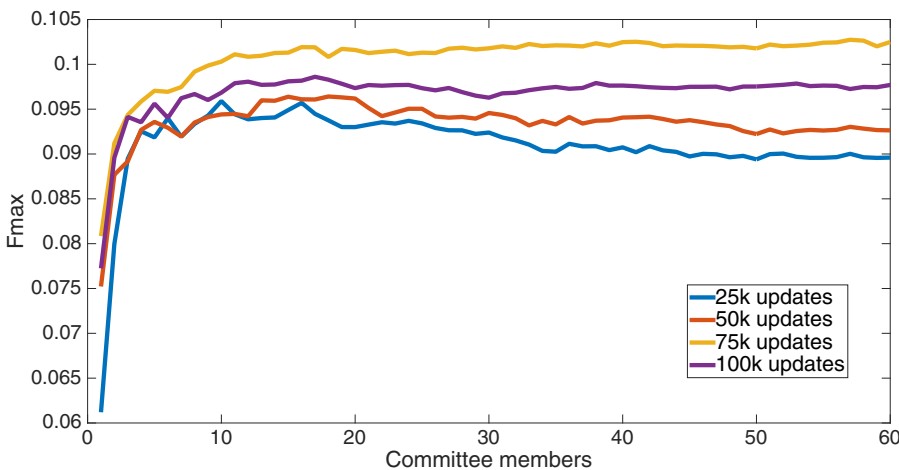

**Figure 4 Treatment results.** Results for the split of 16 with four bits per feature (2,752 total bits). The highest Fmax at 60 committee members is at 75 k, with an Fmax of 0.102.

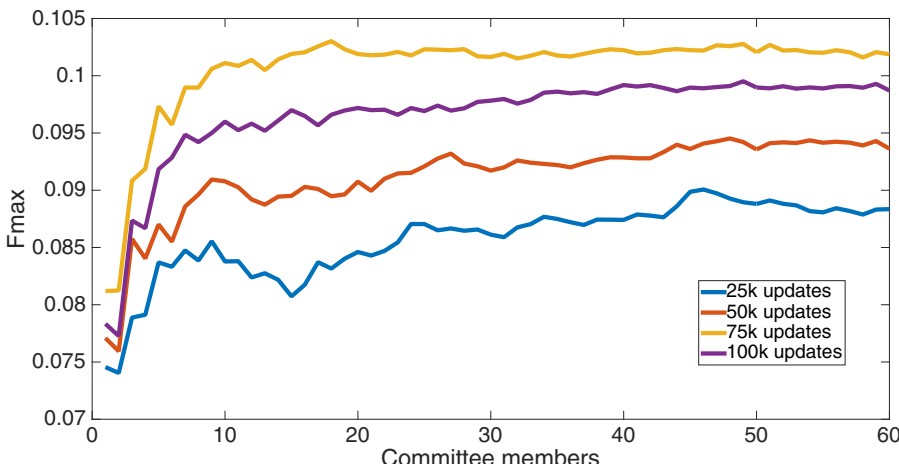

**Figure 5 Treatment results.** Results for the split of four with two bits per feature (1,376 total bits). The highest Fmax at 60 committee members is at 75 k, with an Fmax of 0.102.

each protein and separation length, the number of positive guesses they provided was fixed according to $L$.

As has been noted above, individual Markov networks do not have confidences in their guesses, which are instead only a binary yes or no. However, it is possible to emulate confidence by taking the number of networks in the 60-network committee that provide a positive answer. Using this method as a proxy for confidence, it is possible to provide the top-$L$ positive guesses.

Our specific procedure for providing the top guesses was as follows. For a protein and separation length, we took all positive guesses (defined as those where the majority-vote of Markov networks out of the 60 had given a positive guess). Then, we took all these cases and ordered them by the number of networks that gave a positive answer, taking the top-$L$. If there were ties in confidence that produced more than $L$ guesses, then a random

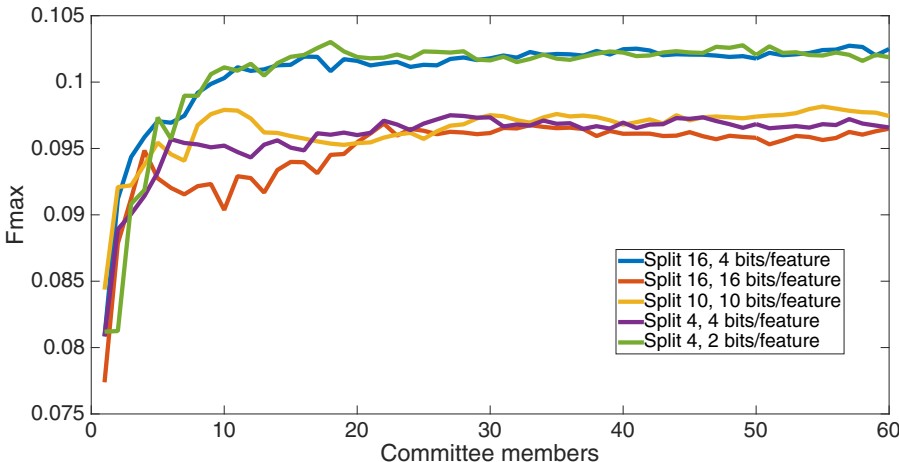

**Figure 6   All treatment results.** Results at 75 k updates for all five split treatments. The highest Fmax is achieved by the split of 16, four bits per feature encoding, with an Fmax of 0.103.

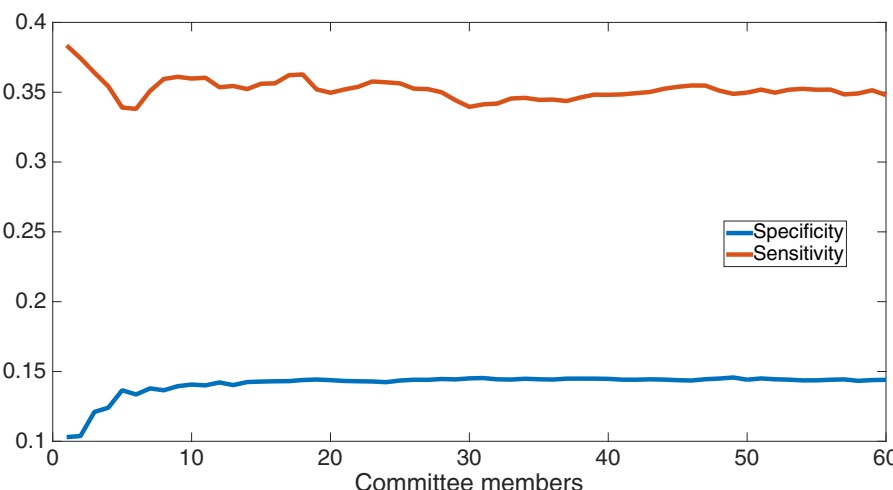

**Figure 7   Treatment results detail.** Specificity and sensitivity results at 75 k updates for the split of four with two bits per feature treatment. Specificity at 60 committee members was 0.14, and sensitivity was 0.35.

selection of the tied examples were taken in order to achieve $L$ guesses. Also, if the number of examples that had a positive answer was less than $L$, then only that amount was taken.

Table 4 gives the total specificity, sensitivity, and Fmax values on the testing set for Markov networks (showing both top-$L$ guesses and all majority-vote guesses) and SVMcon across the several sequence separations. In general, Markov networks do best at short-range and medium-range contacts. Also, at all sequence separations, SVMcon has a higher specificity than the all-guesses and top-$L$-guesses Markov network treatments, but the all-guess Markov network treatment has a higher sensitivity, which is probably because more guesses allows it to cover more true positives. The top-$L$ Markov network treatment has a higher specificity than the all-guess treatment, which makes sense as well, since it is more conservative in its guesses. However, its specificity and sensitivity still lag behind

**Table 4  Markov network (MN) and SVMcon comparison.** Mean specificity, sensitivity, and Fmax of Markov networks (all guesses and top-*L* guesses) and SVMcon across different sequence separations.

| Method | Sep. ≥6 | | | Sep. ≥12 | | | Sep. ≥24 | | |
|---|---|---|---|---|---|---|---|---|---|
| | Spec. | Sen. | Fmax | Spec. | Sen. | Fmax | Spec. | Sen. | Fmax |
| MN, all guesses | 0.144 | 0.349 | 0.102 | 0.132 | 0.281 | 0.090 | 0.108 | 0.209 | 0.071 |
| MN, top-*L* guesses | 0.268 | 0.136 | 0.090 | 0.218 | 0.123 | 0.079 | 0.155 | 0.119 | 0.067 |
| SVMcon | 0.37 | 0.21 | 0.13 | 0.30 | 0.20 | 0.12 | 0.21 | 0.19 | 0.01 |

**Table 5  Markov network (MN) and SVMcon comparison.** Fmax comparisons between Markov network (top-*L* guesses) and SVMcon of different SCOP protein classes and sequence separations.

| SCOP class | Number | Fmax, sep. ≥6 | | Fmax, sep. ≥12 | | Fmax, sep. ≥24 | |
|---|---|---|---|---|---|---|---|
| | | MN, top-*L* | SVMcon | MN, top-*L* | SVMcon | MN, top-*L* | SVMcon |
| Alpha | 11 | 0.075 | 0.120 | 0.012 | 0.087 | 0.009 | 0.050 |
| Beta | 10 | 0.077 | 0.117 | 0.073 | 0.111 | 0.057 | 0.096 |
| Alpha+beta | 15 | 0.102 | 0.161 | 0.089 | 0.146 | 0.067 | 0.110 |
| Alpha/beta | 7 | 0.099 | 0.126 | 0.096 | 0.121 | 0.097 | 0.117 |
| Small | 4 | 0.085 | 0.120 | 0.060 | 0.113 | 0.023 | 0.063 |
| Coil-coil | 1 | 0.091 | 0.142 | 0.030 | 0.025 | N/A | N/A |
| **All** | **48** | **0.090** | **0.134** | **0.079** | **0.120** | **0.067** | **0.010** |

those of SVMcon. These results suggest that our proxy method of taking the answers with the highest number of positive networks is able to at least produce more-accurate positive guesses. However, this still does not produce a higher Fmax than the all-guess Markov network treatment nor SVMcon.

Table 5 shows the Fmax results on the testing set across the six SCOP protein classes covered by the testing set and the three sequence separation thresholds. Markov networks do the best with the alpha+beta and alpha/beta classes. However, Markov Networks do not perform as well as SVMCon, which is in line with Table 4, where the top-*L* guess Markov network treatment did not have Fmax values that were as good as the all-guess Markov network treatment nor SVMcon.

## Performance on CASP 10 and CASP 11 datasets

We next tested our method on target domains available from the Critical Assessment of Techniques for Protein Structure Prediction (CASP) challenges http://www. predictioncenter.org. CASP is a bi-annual event in which groups of researchers test their structure prediction methods on a set of target proteins provided by the CASP organizers. Here, we compared our protein contact map prediction method against selected methods from the CASP 10 *Moult et al. (2014)* and CASP 11 *Moult et al. (2016)* challenges. These methods include CoinDCA (*Ma et al. (2015)*), PSICOV (*Jones et al. (2012)*), plmDCA (*Ekeberg et al. (2013)*), NNcon (*Tegge et al. (2009)*), GREMLIN (*Kamisetty, Ovchinnikov & Baker (2013)*), CMAPpro *Di Lena, Nagata & Baldi (2012)*, and EVfold (*Marks et al. (2011)*). The methods compared are described in detail in the Introduction section.

**Table 6  CASP 10 results comparison.** Specificity results on the 123 CASP 10 targets for short-range, medium-range, and long-range contacts with guess numbers of $L/10$, $L/5$, and $L/2$ where $L$ is the protein length.

| Method | Seq. sep. of [6,12] | | | Seq. sep. of [12,24) | | | Seq. sep. of $>24$ | | |
|---|---|---|---|---|---|---|---|---|---|
| | *L/10* | *L/5* | *L/2* | *L/10* | *L/5* | *L/2* | *L/10* | *L/5* | *L/2* |
| Markov networks | 0.292 | 0.250 | 0.194 | 0.297 | 0.261 | 0.223 | 0.153 | 0.135 | 0.111 |
| CoinDCA | 0.517 | 0.435 | 0.311 | 0.500 | 0.440 | 0.340 | 0.412 | 0.351 | 0.279 |
| PSICOV | 0.234 | 0.191 | 0.140 | 0.310 | 0.259 | 0.192 | 0.276 | 0.225 | 0.168 |
| plmDCA | 0.264 | 0.218 | 0.152 | 0.344 | 0.289 | 0.214 | 0.326 | 0.280 | 0.213 |
| NNcon | 0.499 | 0.399 | 0.275 | 0.393 | 0.334 | 0.226 | 0.239 | 0.188 | 0.001 |
| GREMLIN | 0.256 | 0.212 | 0.161 | 0.343 | 0.280 | 0.229 | 0.320 | 0.278 | 0.159 |
| CMAPpro | 0.437 | 0.368 | 0.253 | 0.414 | 0.363 | 0.276 | 0.336 | 0.297 | 0.227 |
| EVfold | 0.193 | 0.165 | 0.130 | 0.294 | 0.249 | 0.188 | 0.257 | 0.225 | 0.171 |

Instead of evolving another set of Markov networks to test on the CASP targets, we instead tested our evolved networks on the full set of targets. The evolved networks we used were from the best-performing treatment; that is, the set of 60 networks that were from the split of four with two bits per feature at 75 k updates.

The compared methods were the same as those from *Ma et al. (2015)* and the scores used are from that study. The comparison also used the same parameters including the use of the amino acid alpha carbon distances and a distance threshold of eight angstroms. We also used the same 123 protein domain targets (from 95 proteins) for CASP 10 and 105 protein domain targets (from 79 proteins) from CASP 11 for evaluation.

The evaluation in *Ma et al. (2015)* was also different than the one we used on our original dataset. First, their sequence separation thresholds were in the interval [6,12] for short-range contacts, (12,24] for medium-range contacts, and $>24$ for long-range contacts. Second, for each target and separation threshold, they took the top $L/2$, $L/5$, and $L/10$ highest-confidence guesses, where $L$ was the protein length. The results they displayed were the specificity of the methods at these fixed number of guesses.

In the previous section, our method used a simple majority vote in order to make a contact determination. However, in order to provide a better comparison with the other methods in CASP 10 and CASP 11, we also fixed the number of positive Markov network guesses for each separation threshold at $L/2$, $L/5$, and $L/10$ for each target. We did this using the same protocol used when we compared our method to SVMcon; that is, to emulate confidence, we ordered the guesses for each protein by the number of positive network answers our of 60.

Table 6 shows our CASP 10 specificity results in comparison to those shown in *Ma et al. (2015)*, and Table 7 shows the same for CASP 11. Note that for CASP 11, results for NNcon and CMAPpro were not used.

On the CASP 10 dataset, Markov networks do relatively well, performing better than PSICOV, plmDCA, GREMLIN, and EVfold at short range at $L/10$, $L/5$, and $L/2$. At medium range, Markov networks have comparable performance to PSICOV and EVfold at all $L$-values, and do roughly as well as plmDCA, NNcon, and GREMLIN at $L/2$. At long range, Markov networks have difficulty compared to the other methods.

**Table 7  CASP 11 results comparison.** Specificity results on the 105 CASP 11 targets for short-range, medium-range, and long-range contacts with guess numbers of $L/10$, $L/5$, and $L/2$ where $L$ is the protein length.

| Method | Seq. sep. of [6,12] | | | Seq. sep. of (12,24] | | | Seq. sep. of >24 | | |
|---|---|---|---|---|---|---|---|---|---|
| | $L/10$ | $L/5$ | $L/2$ | $L/10$ | $L/5$ | $L/2$ | $L/10$ | $L/5$ | $L/2$ |
| Markov networks | 0.287 | 0.254 | 0.197 | 0.270 | 0.235 | 0.195 | 0.142 | 0.127 | 0.108 |
| CoinDCA | 0.452 | 0.391 | 0.286 | 0.430 | 0.365 | 0.254 | 0.279 | 0.240 | 0.186 |
| PSICOV | 0.190 | 0.144 | 0.112 | 0.196 | 0.163 | 0.115 | 0.198 | 0.172 | 0.127 |
| plmDCA | 0.185 | 0.144 | 0.107 | 0.208 | 0.165 | 0.122 | 0.226 | 0.214 | 0.161 |
| GREMLIN | 0.183 | 0.145 | 0.106 | 0.193 | 0.162 | 0.121 | 0.215 | 0.206 | 0.160 |
| EVfold | 0.159 | 0.137 | 0.100 | 0.197 | 0.163 | 0.113 | 0.193 | 0.163 | 0.132 |

For CASP 11, Markov networks again do better than PSICOV, plmDCA, GREMLIN, and EVfold on short-range contacts, and are also better than these four methods at all $L$-levels for medium-range contacts. Markov networks still have difficulty with long-range contact predictions on the CASP 11 dataset. However, the difference between Markov networks on long-range contacts and the other methods in CASP 10 and CASP 11 is comparable to other performance differences at other ranges and $L$-values between the methods studied.

The respectable performance of Markov networks on the CASP 10 and CASP 11 data indicate that Markov networks may be good at generalizing to other datasets and not just the dataset they are trained on. In addition, it is encouraging that, even though this is the first time that this method has been used, it performs better than others in some cases, such as in short-range and medium-range contact predictions.

## Network recognition of features

In an evolutionary run, Markov networks evolve to use certain input bits (and therefore features) to make their decisions; that is, they evolve to recognize the subset of input bits that give them the best answer ("salient" bits). Figure 8 shows a typical network with its output from the treatment that used a split of four with two bits/feature, at 75,000 updates. Each green circle is an input used by the network, with the red circles representing gates and the blue circles outputs. Because this figure is illustrative, the inputs and gates are unordered. There are several things to notice in this figure. First, even though there are 1,376 possible bits that the network could use, it only connects to 118 in this example. Second, in this network it is common for gates to have connections to multiple input nodes and for outputs to have many gate inputs—the networks are quite complex. Third, there are three outputs. Our system allows up to two outputs to evolve for each class label, so that one output in a pair can serve as a "veto" output to prevent too many "yes" outputs. Here, the positive contact label (on the right) evolved two outputs.

Because of the nature of the evolutionary process, the networks can select the best features in an unbiased manner. Figure 9 demonstrates how networks focus on some features rather than others. In this histogram, taken from the results from the encoding that uses a split of four with two bits/feature at 75 k updates, we plot the frequency distribution of features used over the networks that use them, by assuming that a feature is "recognized" if there is at least one network that reads from one of the bits for that feature.

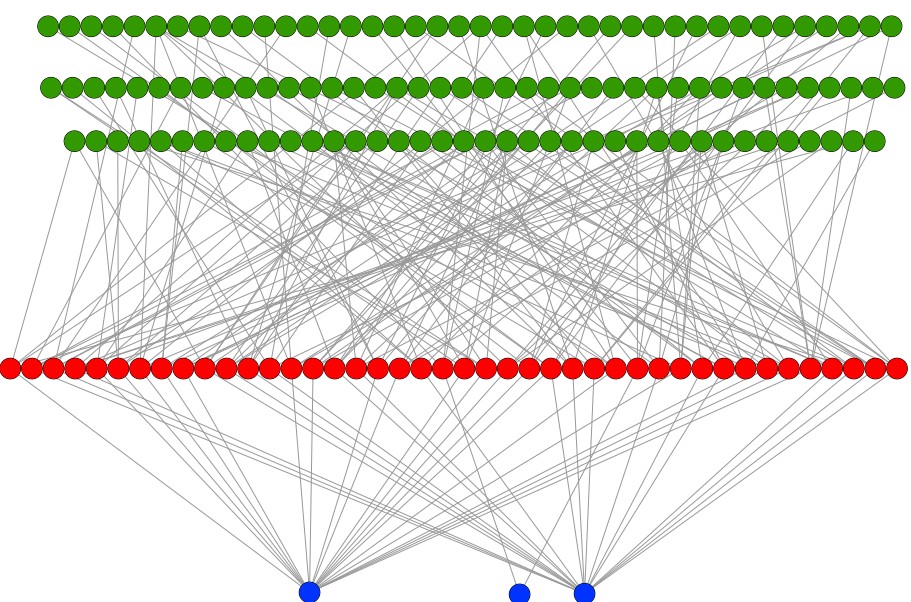

**Figure 8  Sample Markov network.** A sample network diagram taken from the treatment of a split of four with two bits/feature at 75 k updates. Out of a possible 1,376 bits, the network has evolved to recognize only 118 of these. Inputs bits are green, gates are red, and outputs are blue. The inputs and gates are un-ordered. Note that a pair of outputs has evolved to represent a positive contact answer (the maximum is two), but that the negative contact answer evolved only one.

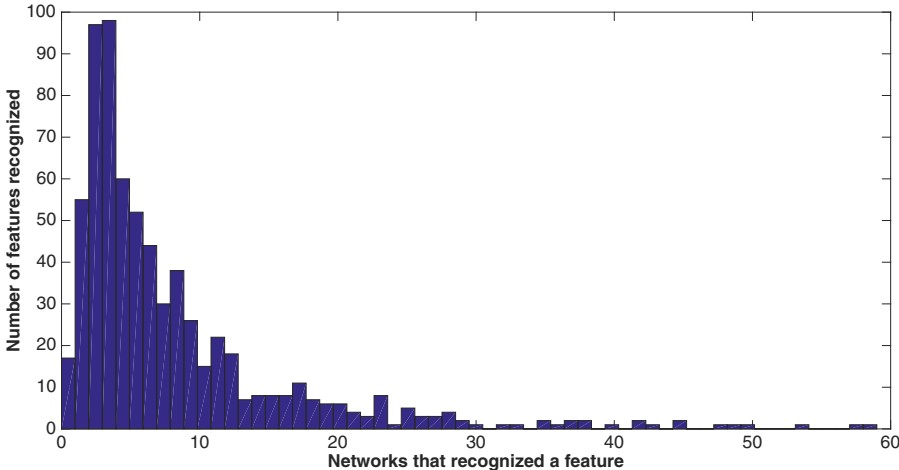

**Figure 9  Network recognition.** A histogram showing how many features are recognized by a certain number of networks (encoding split of four with two bits/feature at 75 k updates). A network only has to have input from one bit of a feature to recognize it.

We see that most features are recognized by only a few networks—for example, nearly a hundred features are recognized by only three networks, with nearly another hundred recognized by only four networks. However, less than 20 features are recognized by no networks. This may be due to networks randomly evolving to recognize a feature before losing that recognition in the evolutionary process, such that it would be very difficult for

**Table 8  Networks per feature.** The mean and median number of networks that recognize each feature for each treatment.

| Treatment | Networks recognized per feature | |
|---|---|---|
| | **Mean networks** | **Median networks** |
| Split 10, 10 bits/feature | 7.5 | 5 |
| Split 16, 16 bits/feature | 7.1 | 4 |
| Split 4, 4 bits/feature | 7.2 | 5 |
| Split 16, 4 bits/feature | 6.5 | 4 |
| Split 4, 2 bits/feature | 7.8 | 5 |

**Table 9  Feature recognition.** Feature recognition values of the networks from each treatment at 75 k updates. For a network to recognize a feature, it only has to connect to one bit for that feature.

| Treatment | Network statistics | | | |
|---|---|---|---|---|
| | **Mean bits recognized** | **Median bits recognized** | **Mean features recognized** | **Median features recognized** |
| Split 10, 10 bits/feature | 98.9 | 98.0 | 85.6 | 84.5 |
| Split 16, 16 bits/feature | 93.5 | 89.5 | 80.9 | 75.5 |
| Split 4, 4 bits/feature | 91.9 | 95.0 | 82.0 | 84.5 |
| Split 16, 4 bits/feature | 84.1 | 85.5 | 74.9 | 76.5 |
| Split 4, 2 bits/feature | 97.6 | 98.0 | 90.0 | 90.5 |

no networks to recognize a feature. Also, no features are recognized by *all* 60 networks, although there are a few that come close.

We note that histograms of the four other treatments at 75 k updates (split of 10, 10 bits/feature; split of 16, 16 bits/feature, split of four, four bits/feature, and split of 16, four bits/feature) were made, but were extremely similar in shape to that of the histogram in Fig. 9. Therefore, they are not shown. However, Table 8 gives for each treatment at 75 k updates the mean and median number of networks that recognized each of the 688 features. As one can see, the numbers across all treatments are very similar.

Table 9 gives information on the general statistics of the networks according to how many bits and features they recognized (used), using all five treatments at 75 k updates. As shown in this table, the mean and median for each treatment tend to be quite similar when examining the number of bits and number of features recognized by the networks. In addition, even though the maximum number of possible bits recognized in each treatment is much higher than the number of features, the number of bits recognized is not very much higher than the number of features recognized (although the number of bits is slightly higher than the number of features). In addition, overall, the mean and median number of features recognized in all treatments is quite small compared to the maximum of 688.

It is desirable to examine more closely which features Markov networks select, because it stands to reason that these salient features are important in general for the task of contact map determination. Table 10 gives a list of the top 12 features in the treatment according to the number of networks that recognized them, using the same treatment of a split of four with two bits/feature at 75 k updates. It is clear from this table that secondary structure and solvent accessibility features are very important to Markov network decisions, and

**Table 10 Most-recognized features.** The 12 features most recognized by Markov networks for the split of 4 with 2 bits/feature at 75 k updates.

| Feature | Networks |
|---|---|
| Contact pair sequence separation $\geq 50$ | 59 |
| C-terminus amino acid window position 5, sheet secondary structure | 58 |
| N-terminus amino acid window position 5, sheet secondary structure | 54 |
| Amino acid central segment window position 4, coil secondary structure | 50 |
| Amino acid central segment window position 4, sheet secondary structure | 49 |
| Contact pair sequence separation $\leq 49$ | 48 |
| Amino acid central segment window position 5, coil secondary structure | 45 |
| N-terminus amino acid window position 5, exposed solvent accessibility | 45 |
| Amino acid central segment window position 5, sheet secondary structure | 43 |
| Contact pair sequence separation of 6 | 42 |
| N-terminus amino acid window position 5, buried solvent accessibility | 42 |
| C-terminus amino acid window position 5, buried solvent accessibility | 40 |

**Table 11 Treatment top features.** Number of top-12 features from the split 4, 2 bit/feature treatment that are also in the top-12 features of the other 4 treatments.

| Number of treatments | Top-12 features |
|---|---|
| Five treatments | 5 |
| Four treatments | 7 |
| Three treatments | 8 |
| Two treatments | 10 |

presumably to determination of contact map prediction in general. The importance of contact pair sequence separation to the networks, specifically a separation of six and a separation of $\geq 50$, seems to suggest that the networks use the extremes of distance as a way to help with classification. Furthermore, these 12 features were also common in the top 12 of the other four treatments. Table 11 shows how many of the top 12 features from the split of four, two bits/feature treatment were also in the top 12 features of the other four treatments. Out of these 12 features, five were in all five treatments, and 10 were in at least one other treatment in addition to the split four, two bits/feature treatment.

Figure 10 demonstrates feature usage related to secondary structure features. Each pair of amino acids in a training or testing sample is situated in a sliding window of size 9, giving a total number of 18 positions. There are a number of features at each of these positions, including the secondary structures. At each of these window positions, the figure shows the number of networks (out of the 60) that evolved to recognize the three kinds of secondary

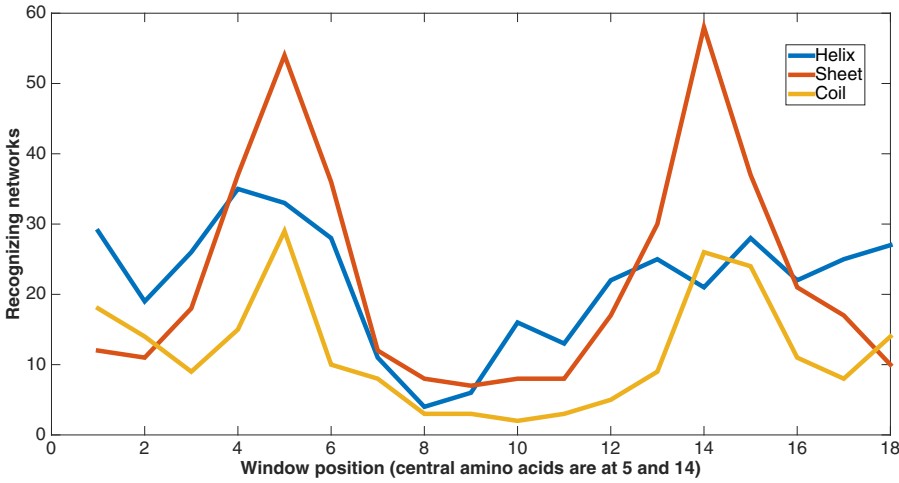

**Figure 10  Secondary structure recognition.** Number of networks out of the 60 that evolved to recognize each kind of secondary structure along the two size-9 sliding windows. Encoding was split of four, two bits/feature.

structure after 75,000 updates. We also see that the most common type of secondary structure recognized by Markov networks is sheet. Further, Markov networks focus on recognizing secondary structures that are closer to the central amino acids. Interestingly, even though the number of networks for each kind of secondary structure decreases as the window positions move away from the center, the number of networks for the "outside" positions for each window is greater for both helix and coil. It is also clear that the peaks in the centers and outsides of the windows are high in absolute terms as well; most features in the dataset are not recognized by many networks.

Similarly, Fig. 11 shows from the same treatment how many networks out of 60 evolve to recognize each feature that describes the sequence separation between the amino acid pair. There is a clear focus on sequence separations that are either very small or very large. One can see that, at least in relation to this dataset, Markov networks are recognizing that certain pair separations are more useful than others.

It is clear from these figures that a number of secondary structure and sequence separation features are important to the networks. Indeed, nine out of the top 10 features in terms of networks that recognized them deal with either secondary structure or sequence separation (the tenth deals with solvent accessibility).

As mentioned before, since Markov networks will tend to evolve to recognize features that benefit them, they will usually only recognize a small subset of the total features. To further demonstrate that the features chosen by Markov networks are useful, other evolutionary runs were performed that had as input only the most-used features in the original runs. To this end, the 60 networks from the encoding that used a split of four with two bits per feature were examined after a run of 75,000 updates to see which features they recognized. The training and testing datasets were changed to contain only features that were recognized by at least six of the networks, and a new evolutionary run with the same parameters was performed on this reduced-feature dataset. The number of features in this

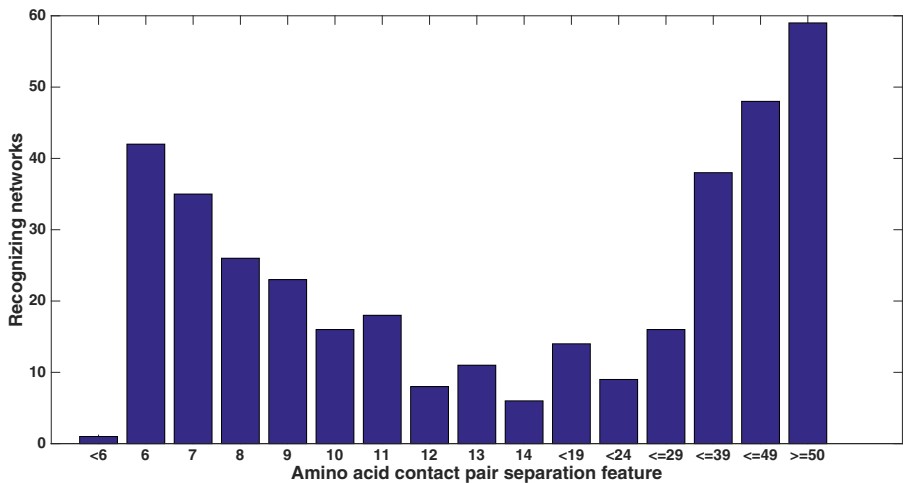

**Figure 11** **Amino acid separation recognition.** Number of networks out of the 60 that evolved to recognize the amino acid pair separation features. Encoding was split of four, two bits/feature. Each tick shown is a different contact separation feature.

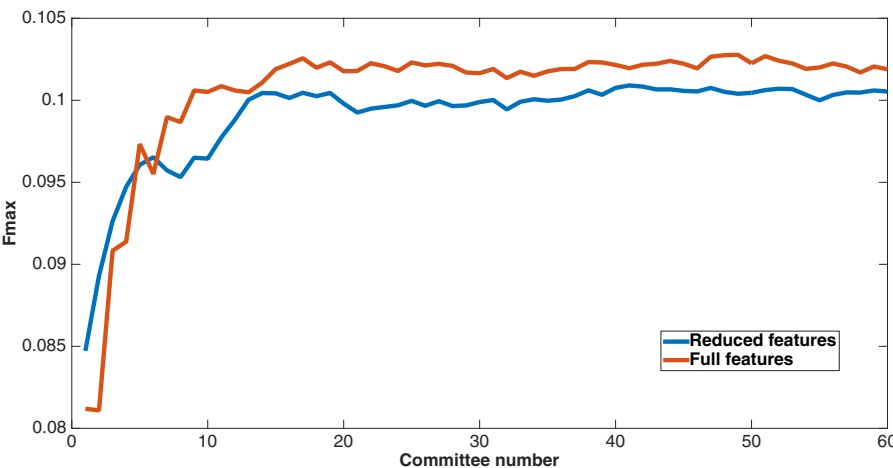

**Figure 12** **Comparison of full *vs.* reduced feature set performance.** Fmax of the original split-4, two bits per feature encoding with all features, and the same kind of run with the reduced feature set that only used features recognized by at least six of the networks from the first run.

new run was 309, or roughly 45 percent of the original number of features. Figure 12 shows the Fmax results of this new run compared to the original run. Although the Fmax results of the new run are smaller, this difference is quite small, indicating that Markov networks have discovered what the salient features for solving the problem. Also, it demonstrates that searching for the right features via trial and error is perhaps not necessary; simply picking the features that the most networks recognize is suitable for finding the best features.

This figure demonstrates two things. First, many features that were chosen for the dataset were unncessary to obtain the same level of performance. In addition, Markov networks successfully evolved to discover which features were useful in contact map prediction.

It is worth noting that many of the 688 features that were not used in the reduced feature set were features that described the amino acid percentages that were based on a sequence alignment of the proteins. Considering the two size-9 windows and the size-5 central segment window, and the fact that the amino acids were the 20 canonical amino acids plus a gap, there were 23*21 = 483 total features in the dataset based on amino acid percentages alone. Yet only 164 of these (34 percent) were used in the reduced-feature dataset, as opposed to 145 of the remaining 205 features (71 percent of the remaining). Some notable observations included the fact that all but one (22 out of 23) of the gap amino acid features qualified. This could be due to the importance of gap additions in sequence alignments. In addition, most of the central segment window amino acid features were salient—88 out of 104, or 85 percent—and often had a relatively high number of networks that recognized them.

The salient features of the size-9 windows around the contact pairs were not as numerous (76 out of 378 features, or 20 percent) and tended to have relatively low numbers of networks that recognized them. An interesting exception was the feature for cysteine in the fifth position (central position) for the C-terminus window. A total of 19 networks recognized this feature. Because cysteine is so important to protein structure, this is not surprising, and the cysteine feature in the center of the N-terminus window was recognized by eight networks as well.

## DISCUSSION

As demonstrated in our results, the performance of the evolution of Markov networks depends mainly on two things: First, the fitness function used, and second, the encoding of the dataset. For example, with respect to encoding, the two treatments that performed the best were the two that used the reduced-bit binary encodings. It is hypothesized that one reason that these two treatments did better than the other three is that there were fewer bits for Markov networks to evolve to choose, but also that condensing bits would "force" Markov networks to evolve over all bits for a continuous feature due to the nature of the binary encodings.

Also, with respect to fitness functions, the best found so far has been accuracy. This is perhaps due to the fact that accuracy is such a simple fitness function—it is simply the proportion of correct guesses (true negative and true positive) in the dataset and does not require a complex formula. Furthermore, while other fitness functions such as Fmax measure, specificity/sensitivity, or Matthews correlation coefficient (*Matthews, 1975*) have been tried and have not performed as well as accuracy, there is the intriguing idea that a "committee of committees" could be used based on a combination of the answers from runs of several fitness functions. Furthermore, it has been noted that a different fitness function/encoding combination might produce better results; in addition, a different type of evolutionary algorithm (such as NSGA2 (*Deb et al., 2002*)) or different evolutionary parameters, such as the type of population replacement (in our case, we used tournament selection) (*Blickle & Thiele, 1996*) could possibly foster a more-productive fitness landscape and therefore performance.

There are several comments to be made regarding the comparison of Markov networks to other methods. First, it is somewhat disappointing that Markov networks (both all-guess and *L*-guess treatments) did not do as well as SVMcon at the different separation ranges and protein classes, although the all-guess Markov network treatment did better in terms of sensitivity, as expected. In particular, the Fmax of the *L*-guess treatment for Markov networks was lower than that of the all-guess treatment. Although using the number of positive networks as a proxy for confidence did seem to work somewhat, it could be that it was not quite as helpful as the confidence levels of other methods might be.

It was more encouraging that the evolved Markov networks were able to perform reasonably well on the CASP 10 and CASP 11 datasets, performing as well or better than several methods on short-range and medium-range contacts. This could indicate that Markov networks can be useful in generalizing to datasets that are much different to those that they are trained on. It is also encouraging due to the fact that this is the first time the evolution of logic circuits (i.e., Markov networks), has been used on this problem.

The open-ended nature of evolutionary computation is both a blessing and a curse. The usage and theory of evolutionary computation is continuously being worked on and improved by many researchers, allowing for the possibility of a great increase in performance of this method. However, it is always unknown which specific fitness function and other parameters to use, and thus there is still an element of trial and error.

One of the primary strengths of the evolution of Markov networks is that the evolution proceeds in an unbiased manner. Thus, as demonstrated in the results section, networks evolve to recognize some features more than others. This is useful, since it allows one to differentiate some features as being "better" than others according to how many networks evolve to recognize them. Figure 9 shows that there is quite a bit of diversity in terms of which features are recognized as salient, and Table 10 shows how certain groups of related features features (e.g., secondary structure and solvent accessibility) are important to the decisions of the networks. Furthermore, Table 9 shows that the networks are fairly parsimonious in their decisions and need only a relatively small fraction of the total bits to make their decisions. Finally, Fig. 12 shows that, when considering only the most-used features, they are capable of performing almost as well as when using all the features. This parsimonious behavior that focuses on only a few salient features, and which can also identify groups of related features, shows that Markov networks do not necessarily have to grow to an arbitrary and unwieldy complexity in order to achieve improved results, even though they are capable of doing so.

## CONCLUSION

We have shown that evolution is able to produce Markov networks that are able to make contact map predictions and recognize relevant features from the SVMcon dataset. As far as we know, this is the first time that such a method has been used on a bioinformatics problem. The results show that the method is promising, and may have wider applications. Naturally, because this is the first time such a method has been used in this manner, there is room for improvement. There are a number of specific ways in which our method

could be improved, including using a different evolutionary algorithm, fitness function, Markov network structure, encoding, or feature space. We could also use our method in conjunction with other methods; for example, if used with other methods, our method and others could form a committee where each receives a voting weight for a contact pair example. One such additional method could include PSICOV (*Jones et al., 2012*), which involves finding correlated mutations between a protein containing the contact pairs and a large database of proteins. Indeed, this idea has been used with PSICOV itself in the form of MetaPSICOV, which uses an ensemble of correlated-mutation methods in a committee *Jones et al. (2015)*. Other possible additional methods include the LASSO feature selection method (*Tibshirani, 1996*), and random forests (*Liaw & Wiener, 2002*; *Rainforth & Wood, 2015*), among others.

Furthermore, we have shown that the evolution of Markov networks can, in an unbiased manner, produce networks that recognize relevant (salient), useful features from a dataset. Choosing these salient features could help to remove extraneous features, a task that would otherwise be computationally intensive. This could be especially relevant to a problem that uses many more features (in the thousands or tens of thousands) than ours. Also, one could use this feature information as a prediction tool to guess which kinds of features are useful or not. For example, if it is demonstrated that all features relating to a particular class of amino acid (e.g., polar amino acids) are used by Markov networks, then it might be desirable to find more features of this type. Furthermore, this offers the possibility that Markov networks could be used as a general-purpose feature detector in other scientific work.

## ACKNOWLEDGEMENTS

We wish to thank Dr. David B. Knoester for developing the Markov network codebase used here and for help adapting it to this work. Part of the work was performed using XSEDE resources as well as computational resources at Michigan State University.

### Funding

This material is based in part upon work supported by the National Science Foundation under Cooperative Agreement No. DBI-0939454 (BEACON Center) and National Science Foundation under Grant No. 1647884. There was no additional external funding received for this study. The funders had no role in study design, data collection and analysis, decision to publish, or preparation of the manuscript.

### Grant Disclosures

The following grant information was disclosed by the authors:
National Science Foundation: DBI-0939454, 1647884.

### Competing Interests

Claus O. Wilke is an Academic Editor for PeerJ.

## Author Contributions

- Samuel D. Chapman performed the experiments, analyzed the data, contributed reagents/materials/analysis tools, wrote the paper, prepared figures and/or tables, reviewed drafts of the paper.
- Christoph Adami and Claus O. Wilke conceived and designed the experiments, contributed reagents/materials/analysis tools, wrote the paper, reviewed drafts of the paper.
- Dukka B KC conceived and designed the experiments, performed the experiments, analyzed the data, contributed reagents/materials/analysis tools, wrote the paper, reviewed drafts of the paper.

## Data Availability

Chapman, Samuel (2016): Output results and testing contacts. figshare. Available at https://doi.org/10.6084/m9.figshare.3463256.v1.

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
