# Peer review of "The evolution of logic circuits for the purpose of protein contact map prediction"

_PeerJ, doi:10.7717/peerj.3139_

## Round 0.1 · original submission · Major Revisions

Your manuscript has been reviewed by two experts in the field. As you will find in their comments, cited below, both of them raise substantial criticisms. One of them even suggests its rejection. In particular, your method is not well compared to other methods, some of which are published rather recently. If you read their comments carefully and think that you can address all of them, please revise your manuscript following their comments; otherwise, please you may want to reconsider publishing it this time.

·

Basic reporting

Although the authors mention (albeit too lightly) on the PSICOV method by Jones et al., they almost completely ignore the recent breakthroughs in the field of protein contact prediction such as Morcos et al. (2011) (http://www.pnas.org/content/108/49/E1293), Miyazawa (2013) (http://journals.plos.org/plosone/article?id=10.1371%2Fjournal.pone.0054252) and the works that followed thereafter. These recent methods based on the analysis of pairwise correlations in multiple sequence alignments literally changed the landscape of the field, and it is a serious flaw of the authors not to consider these works.

Experimental design

The authors use the Fmax-value as the only measure of performance, which is not a standard measure in contact prediction. In other literatures of protein contact prediction, performance measures such as the true-positive rate (precision) or area under the ROC curve (AUC) are used. Therefore, it is not easy to compare the present work with others, which makes the manuscript less meaningful.

Validity of the findings

Acceptable.

Reviewer 2 ·

Basic reporting

This article applied Markov network to the prediction of contact map of protein structures. This is a new trial of this method to biological fields. I think this work has two large problems to be accepted.

Experimental design

The methods are not detailed described. For example, no information on how many and what kind of protein structures used, how to reduce contact samples, what are 688 features (621 are described, but the others have no description), and no “contact” cutoff (just 8A, no specific atom) are given. Although the method seems to create alignments, I could not find how to make them and how these alignments are used. The text cites the SVMcon article for the details of the dataset, however, an article have to explain what the authors did in the manuscript.

Validity of the findings

Although this is new trial, the results are not superior to the pre-existing program. Even if it does not out-perform with the pre-existing programs, it should be at least comparable with others to be accepted.

Additional comments

The article compared its performance only with SVMcon. There are many programs and they should be referred to validate the programs, for example, Pollastri et al., Proteins, 47, 228 2002; Tegge et al., NAR, 37, W515, 2009; Lena et al., Bioinformatics, 28, 2449 2012.

---

## Round 0.2 · Minor Revisions

Your revised manuscript has been reviewed by the two original referees. Both of them basically agree that the manuscript is now almost acceptable but one of them points out several minor points, as shown below. Please check them and re-revise them if you feel that they are reasonable.

·

Basic reporting

English is clear enough. Now references to and comparison with recent contact prediction papers are included. Although the presented method is not the best among all that are compared, I think this manuscript contains useful information on contact prediction.

Experimental design

I think it is sound.

Validity of the findings

This paper contains some useful information on evolution of Markov networks (MNs) as well as its application to contact prediction. Many of the recent methods are compared with the authors'. While the presented method is not the one best among those compared, especially for long-range contacts, this paper may serve as a good reference work for applications of the evolutionary approach to learning MNs.

Reviewer 2 ·

Basic reporting

no comment

Experimental design

no comment

Validity of the findings

no comment

Additional comments

The authors answered all of the comments raised for the previous version. This version describes the details of the dataset in terms of protein structures, which is acceptable to see the summary of the dataset. They also performed the comparisons with other methods but SVMcon with the use of CASP10 and 11. Although this method does not clearly outperform other programs, it is comparable or, in some cases, outperform others. By taking the first application of the logic circuits for this field, I judged that this version is acceptable for the publication in PerrJ with the minor points listed below.

1) The “Network Recognition of Features” section is intriguing. However, it is not mentioned which type of input is used except for Fig. 8 (“split of 4 with 2 bits” in line 434). From the context, I guessed that Fig.9, Table8, and Table 9 use the same input. It is better to clearly describe it.
2) Related with the comment above, are the results same by the other inputs? Regardless of same or not, it is better to include a relation between split types and recognized features.
3) In the text, “Markov network”, “the Markov network”, “MNs”, and “the MNs” are used. One choice of them must make the paper easily read.
4) The section of the line 234 – 239 and that of the line 240 – 248 may be a repeat.
5) In the line 247, I could not understand “input used 10 bits”. More explanation may be needed.
6) In the line 260, “16*11008” must be a typo.

---

## Round 0.3 · accepted · Accept

I am pleased to inform you that I find your re-revisions reasonable and thus decide its acceptance.